# Food Insecurity Is Associated with the Quality of Diet of Non-Institutionalized Older Adults from a Southern Chilean Commune: A Cross-Sectional Study

**DOI:** 10.3390/nu14010036

**Published:** 2021-12-23

**Authors:** Orietta Segura-Badilla, Ashuin Kammar-García, Addí Rhode Navarro-Cruz, Jacqueline Araneda-Flores, Javier Mancilla-Galindo, Obdulia Vera-López, Martin Lazcano-Hernández, Gladys Quezada-Figueroa

**Affiliations:** 1Programa UBB Saludable, Facultad de Ciencias de la Salud y de los Alimentos, Universidad del Bío-Bío, Chillan 3780000, Chile; osegura@ubiobio.cl; 2Dirección de Investigación, Instituto Nacional de Geriatría, Mexico City 10200, Mexico; akammar@inger.gob.mx; 3Sección de Estudios de Posgrado e Investigación, Escuela Superior de Medicina, Instituto Politécnico Nacional, Mexico City 11340, Mexico; 4Departamento de Bioquímica y Alimentos, Facultad de Ciencias Químicas, Benemérita Universidad Autónoma de Puebla, Puebla 72570, Mexico; addi.navarro@correo.buap.mx (A.R.N.-C.); obdulia.vera@correo.buap.mx (O.V.-L.); lazmar@gmail.com (M.L.-H.); 5Departamento de Nutrición y Salud Publica, Facultad de Ciencias de la Salud y de los Alimentos, Universidad del Bío-Bío, Chillan 3780000, Chile; jaraneda@ubiobio.cl; 6Unidad de Investigación UNAM-INC, Instituto Nacional de Cardiología Ignacio Chávez, Mexico City 14080, Mexico; javimangal@gmail.com

**Keywords:** older adults, healthy eating index, food consumption, nutrition of older adults

## Abstract

As the population ages, greater attention to age-related health problems related to diet and lifestyles is needed. Here, we sought to evaluate the associations between demographic and clinical characteristics and food insecurity with the quality of diet of non-institutionalized elderly from a southern Chilean commune. We performed an analytical cross-sectional study in a sample of 376 older adults. Nutritional status was evaluated through anthropometric measurements. Quality of diet was determined by the healthy eating index (HEI), obtained through the frequency of consumption questionnaire. Socioeconomic, demographic, and lifestyle variables were also collected. Ordinal logistic and Poisson regression models were applied to study associations with quality of diet. The sample consisted of more women (81.6%) than men (18.4%). Most older adults were found to live in a situation of vulnerability or poverty (82.4%), with most having food security (65.7%). According to the HEI, only 14.1% had a good quality of diet, 83.8% had diet in need of improvement, and 2.1% had an unhealthy diet. There was an association of food insecurity and cardiovascular risk (according to waist circumference) with lower quality of diet categories. However, an association with the unhealthy quality of diet category was not confirmed with Poisson regression analysis, which was possibly due to the low number of subjects in that category (n = 8, 2.1%). Other modifiable factors like physical activity, hours of sleep, and polypharmacy were not associated with lower quality of diet categories. Socioeconomic status, which is a structural health determinant, was not associated with decreased quality of diet. Since this was a cross-sectional study performed on a small sample from a Chilean commune, directionality of associations cannot be discerned, and future longitudinal studies could aim to better characterize these associations in larger samples of elderly patients.

## 1. Introduction

The increase in life expectancy has been one of the great achievements of humanity in this century. Life expectancy in Chile reached 80.04 years in 2018 [1]. However, an increase in lifespan is not necessarily accompanied by a better quality of life [2,3].

Old age is not a well-defined stage; the World Health Organization defines it as 60 years or more in developing countries, and as ≥65 years for developed countries, a classification that involves the economic and social situation of countries [4,5]. The elderly are characterized by having high rates of low income, decreased appetite, difficulty buying and preparing food [6], loneliness, dysphagia, and suboptimal nutritional states (8 out of 10 elderly have excess malnutrition) [7].

It can be stated that aging is a biological phenomenon that carries with it certain characteristic changes of advanced age [8]. Each stage of life has its own nutritional needs that respond to different biological, functional, cultural, and social characteristics. Therefore, with aging, it is necessary to adapt the diet to nutritional needs based on the specific situation of age, health status, and capacity [9,10]. Without doubt, the composition of diet influences the course of aging. This influence is clearly reflected by the role of nutrition in the development of age-related diseases like osteoporosis, type 2 diabetes, cardiovascular disease, and cancer [11,12,13].

Quality of diet and its relationship with health is a key challenge in nutritional epidemiology to detect nutritional problems. There are indicators or indices built from algorithms which are used to categorize how healthy the eating pattern is in the context of behaviors and eating habits of individuals [14]. The existing indices were developed based on dietary recommendations and guidelines for specific places or countries [15]. These indica can be based on nutrients, foods, or food groups, and can be combined into other indices. On this basis, there are 4 widely referenced and validated original indices: the Healthy Eating Index (HEI), the Diet Quality Index (DQI), the Healthy Diet Indicator (HDI) and the Mediterranean Diet Score (MDS) [16]. From these indicators, various adaptations are originated according to dietary and nutritional recommendations for each country. In Chile, there are few studies that assess the global quality of diet in older adults and the existing ones have adopted the methodology proposed by Kennedy [17], to assess the diet of schoolchildren and adults [18,19,20]. Therefore, the objective of this study was to evaluate the associations between demographic and clinical characteristics, and food insecurity with the quality of diet of non-institutionalized elderly from a southern Chilean commune.

## 2. Material and Methods

### 2.1. Study Design and Participants

This cross-sectional analytical study was carried out during the April–July, 2019 period in the Chilean commune of Chillán Viejo, where ~1700 non-institutionalized adults older than 60 years live. The sample size was calculated for the estimation of a proportion by considering the prevalence of food insecurity of 40.4% in older adults living in Chilean communes [21], with 95% significance and 5% precision, resulting in 369 participants. Adjustment for finite samples was used with the formula $n_{a}=n/\left[1+\left(n/N\right)\right]$ in which n is the sample size, and N the size of the population (N = 1700). This resulted in an adjusted sample size of 303 participants. A second adjustment for an estimated 20% exclusions after data collection was performed, resulting in a final sample size of 364 participants. The participants were part of the project “Food Insecurity and Quality of Diet of the elderly from a commune in southern Chile” and an open invitation for this study was made to them. 

The inclusion criteria were men and women with an age >60 years, normal cognitive function according to the abbreviated Mini-Mental State Examination (MMSE) test (≥14 points), and those who provided their written informed consent to participate. Exclusion criteria were the presence of uncontrolled eating disorders and participants with incomplete information.

The study was approved by the Bioethics and Biosafety Committee of the Universidad del Bío-Bío (*DIUBB 186620 2/I*).

### 2.2. Data Collection and Measurements

All data were collected by a team composed of Nutritional Scientists and last-year students of the bachelor’s degree program in nutrition and dietetics, who received training on the standardization of anthropometric measurement collection and surveying; a Standardized Operating Procedures protocol was built for this. A survey was created for the purpose of this study, including 4 sections: (1) personal and sociodemographic data, (2) food insecurity [22,23], (3) nutritional status (anthropometric measurements), and (4) food consumption frequency questionnaire [24]. These four individual sections are described in subheadings bellow. The survey consisted of a total of 50 items within 7 sections (full questionnaire available in the Appendix A).

The survey was validated by 10 experts in nutrition and public health in a pilot test, which consisted of the invitation of 30 persons older than 60 years from the commune, who were subsequently ineligible to be included in the sample of the main study and thus were not included for analyses, who provided their written informed consent to participate in the pilot study to validate the survey. The procedures of the validation involved linguistic and cultural adaptations and viability, by assessing the time employed in the application of the survey, easiness of the format, and the brevity and clarity of the questions. Items were adapted to include language understandable by the elderly, as well as to improve both the way of delivering questions by interviewers and the registration and codification of responses.

The survey was individually applied by previously trained personnel. All the questions were read to participants in the presence of a family member, when available.

#### 2.2.1. Personal and Sociodemographic Data

Personal and sociodemographic data collected were age, sex, number of persons with whom living place is shared, household income, socioeconomic status, medications consumed, hours of sleep, and physical activity. Socioeconomic level was calculated from combinations between education and occupation of the main income provider within household, number of household members, and total income per average month. The classification of the Association of Market Researchers (AIM in Spanish) [25] was used, where: Upper Class (AB) has a median household income of $4386 USD; Wealthy Middle Class (C1a), $2070 USD; Emerging Middle Class (C1b), $1374 USD; Typical Middle Class (C2), $810 USD; Lower Middle Class (C3), $503 USD; Vulnerable Class (D), $307 USD; and Poor Class (E), $158 USD [25]. 

#### 2.2.2. Sleep and Physical Activity

Recommendations for the duration of nocturnal sleep in the elderly from the National Sleep Foundation were used [26] to classify participants into three categories: below recommended (<5 h), recommended (5–7 h), and above recommended (>8 h). Physical activity was considered as the performance of any type of scheduled activity—such as walking, dancing, jogging, yoga, or others—performed outdoors for more than 30 min, more than 3 times a week [27]. Participants reporting full, occasional, or null compliance with this recommendation were classified as “physically active”, “irregularly active”, or “sedentary”, respectively. 

#### 2.2.3. Food Insecurity Assessment

Food insecurity refers to the limited/uncertain availability of nutritionally adequate and innocuous foods or the limited/uncertain ability to obtain them through socially acceptable means [22]. Food insecurity was assessed with the Household Food Insecurity Access Scale (HFIAS) [23], which includes nine questions about household food security in the last four weeks. These items assess worry related to obtention of foods, inability to consume desired foods, substitution for undesired foods, and limited variation of foods, among others. Responses to these items were used to classify respondents into four categories: food security (few or no worries about insufficient food at home), mild food insecurity (concerns about not having enough food, which influences the types of foods consumed), moderate food insecurity (in addition to the prior, they were forced to consume unwanted food), and severe food insecurity (feeling hungry but not eating, or not eating for an entire day, due to lack of money or other resources) [23]. A second variable was created by dichotomizing food insecurity (the categories of mild, moderate, and severe food insecurity were merged) and food security.

#### 2.2.4. Nutritional Status by Anthropometry

Nutritional status was assessed by measuring weight, height, and waist circumference, using standardized anthropometric techniques. The weight in kg was obtained with a floor scale (SECA mod 714, Hamburg, Germany) with a precision of 100 g. For the determination of height in meters (m), a stadiometer (SECA mod 210, Hamburg, Germany) with a precision of 0.1 mm was used. Waist circumference (WC) was measured with a tape measure (SECA mod 201, Hamburg, Germany) placed above the upper border of iliac crests (approximately at the level of the navel). WC measurements were recorded in centimeters and classified as: female cardiovascular risk ≥ 88 cm; male cardiovascular risk ≥ 102 cm, which are the values considered in the ATPIII-NCEP 2001 definition of Metabolic Syndrome. The body mass index (BMI) was calculated from weight and height (kg/m^2^) and the following categories recommended for the elderly: underweight (<22.0 kg/m^2^), normal weight (22.0–26.9 kg/m^2^), overweight (27.0–29.9 kg/m^2^), and obesity (≥30 kg/m^2^) [28]. 

### 2.3. Food Consumption

Data were collected using the food consumption frequency questionnaire [24], since this questionnaire is used as part of the healthy eating index evaluation (see the following subheading). The food groups were: (1) cereals and derivatives, (2) vegetables, (3) fruit, (4) milk and dairy products, (5) fats and oils, (6) meat and fish, (7) legumes and nuts, (8) sausages and cold cuts, (9) sweets and pastry products, and (10) others (sugary drinks, snacks and fast food). For each of the reported food groups, the frequency of consumption was evaluated and categorized as: daily (≥1 time/day), weekly (≥4 times/day), occasional (≥3 times/month), and never or almost never (≤1 time/month). The amount of food consumption was classified according to the number of portions consumed from each food group into the following 3 categories: 1–3 portions, 3–5 portions, and >5 portions. Portions were calculated according to equivalences shown in Appendix A [24,29].

### 2.4. Quality of Diet

To determine the quality of diet, the Healthy Eating Index (HEI) was used [30]. We adapted the questionnaire validated for the Spanish population according to food groups known by the Chilean people, and subsequently validated it in a pilot test, as previously described [31]. Using the frequencies of food consumption, the first 5 food groups were considered for daily consumption; groups 6 and 7 were considered for weekly consumption; and groups 8–10 were considered for occasional consumption. Each of these frequencies was assigned a score between 0 and 10, according to criteria shown in Appendix A. HEI was calculated by adding the score obtained for all variables, which allows a theoretical maximum of 100 points, where 80–100 points corresponds to healthy eating; 50–80 points corresponds to diet in need of improvement; and HEI < 50 points corresponds to unhealthy eating.

### 2.5. Statistical Analysis

The quantitative descriptive data are presented as means with standard deviation, and qualitative descriptive data are presented as frequency and percentage. For comparisons between HEI groups and quantitative variables, a one-way analysis of variance (ANOVA) was applied. For the variables that did not meet the assumption of homoscedasticity, Welch’s ANOVA was applied. For comparisons of qualitative variables, the chi-square analysis of linear trend was used.

Two regression models were applied to determine the degree of association of different anthropometric, demographic, or food consumption variables with quality of diet. The variables introduced into the model were sex, age (numerical scale), BMI (numerical scale), waist circumference (numerical scale), cardiovascular risk, Food Safety Scale (numerical scale), vulnerable or poor socioeconomic status, obesity (BMI ≥ 30), food insecurity, drugs, hours of sleep, and physical activity. The first regression model consisted of an ordinal logistic regression analysis considering the worst quality of diet category according to HEI as the greater ordinal category. The results of this model are presented as the regression coefficient (β), standard error, and 95% confidence interval (95% CI). The second was a Poisson regression model considering a HEI score < 50 as the outcome which corresponds to unhealthy eating. The results of this model are presented as prevalence ratios (PR) and 95% confidence intervals (95% CI). The variables were introduced by the enter method for both models. The statistical assumptions of each model were verified by residual analysis.

All analyses were performed in SPSS v.21 software. A value of *p* < 0.05 was considered a statistically significant difference.

## 3. Results

Out of 402 participants who responded to the invitation and were assessed for eligibility, 376 subjects were included for analysis, with a mean age of 73.5 (SD: 6.9) years. The descriptive data of the sample are shown in Table 1. Most of the subjects were women (81.6%, n = 307). Regarding socioeconomic level, 82.4% (n = 310) of the subjects were in a category of vulnerability (D) or poverty (E). Despite this, most subjects had food security (65.7%, n = 247), whereas 33.2% (n = 125) had some level of food insecurity. Only 14.1% (n = 53) of subjects had a good quality of diet, being classified as healthy eating according to the HEI, while 85.9% (n = 323) had unhealthy eating or the need to be improved. Most of the subjects had healthy habits such as more than 5 h of sleep (89.1%, n = 325), or being physically active (55.9%, n = 210). Despite this, 77.8 (n = 292) of participants were overweight or obese. Most of the subjects had polypharmacy, with an intake of more than 3 drugs per day (68.3%, n = 257). To determine possible differences in sociodemographic and personal characteristics, these were compared with the HEI classification groups, as shown in Table 2.

Table 3 and Table 4 show the comparisons of the food frequencies and food portions consumed among the different HEI groups. Cereals, vegetables, fruits, and fats were consumed more frequently by those with a healthy diet. Consumption of fats was infrequent in those with an unhealthy diet. Dairy foods were the most frequent source of protein on a daily basis, while meats and legumes had a weekly consumption pattern. 

The results from ordinal logistic regression analyses showed that food insecurity and cardiovascular risk were the variables most strongly associated with a low quality of diet (Table 5). The results of the Poisson regression analyses suggest that none of the sociodemographic or anthropometric factors influenced the quality of diet in the elderly (Table 6). 

## 4. Discussion

In this study, we sought to evaluate the associations between demographic and clinical characteristics and food insecurity with the quality of diet of non-institutionalized elderly from a southern Chilean commune. We found that most participants had a poor quality of diet, either with need of improvement (83.8%) or unhealthy eating (2.1%). Food insecurity and cardiovascular risk according to weight circumference were associated with worsening dietary habits in the ordinal logistic regression analysis. None of the variables showed association directly with the unhealthy diet category in the Poisson regression model; the absence of significance with an unhealthy diet could have been due to the low number of subjects in that category (n = 8, 2.1%), which suggests that the deterioration of quality of diet was not severe enough to classify subjects with an unhealthy diet, but instead with the need for improvement; this is one reason why it could be feasible to perform interventions to improve the quality of diet. 

The main importance of these findings is that structural health determinants which are difficult to change through simple interventions like socioeconomic status known to affect quality of diet in younger adults [32] were not associated with quality of diet in our sample of elderly participants. Conversely, food insecurity can be readily addressed through nutritional interventions in diverse populations [33,34], which is one reason why our findings show that it could be worth evaluating the benefits of performing a nutritional community intervention to improve food insecurity, to study if doing so could improve quality of diet in the elderly.

In our sample, 82.4% of participants were found in a category of vulnerability or poverty. Despite this, more than half (65.7%) had food security. Due to the high proportion of participants with low socioeconomic status and suboptimal quality of diet, it could have been expected that these variables would have explained each other. However, after logistic regression analyses, food insecurity was associated with a diet in need of improvement or of poor quality, but not socioeconomic status. Nonetheless, poverty has been long considered an underlying cause of food insecurity, especially under circumstances of extreme poverty and malnutrition [35]. The population with the lowest socioeconomic status consumes significantly lower amounts of fruits, vegetables, and dairy foods that those in higher socioeconomic levels [36]. Since our sample of older adults was not characterized by extreme poverty, this could explain why there was not a strong association between lower socioeconomic status and quality of diet, whereas food insecurity did have an association with a lower quality of diet. 

There are few studies evaluating HEI in samples composed solely of older people, which have shown a high prevalence of an unhealthy quality of diet or in need of improvement according to HEI scores [37]. In Mexican institutionalized older adults, a mean HEI score of 73 was reported [38], whereas in the US, the 2015 WWEIA/NHANES study showed that people over 65 years had an average HEI score of 64 [39]. In Spain, the mean HEI score was 77.2, and 89.6% of participants older than 65 years had a diet in need of improvement [7,38], which was comparable to our findings of a mean HEI = 79.1, and a higher proportion of older adults was found with a quality of diet in need of improvement (83.8%).

A high prevalence of food insecurity was only observed in the group of subjects with an HEI classification of diet in need of improvement (*p* = 0.02). As discussed earlier, food insecurity was associated with a trend towards lower quality of diet categories. It has been proposed also that people in food insecurity compensate for the lack of quality and variety with the palatability that they find in foods with low nutrient content but high in energy density [40,41]. The ELAN study (Latin American study of nutrition and health) demonstrated a large contribution of refined carbohydrates, foods and beverages rich in fats and sugars, and a limited intake of complex carbohydrates and fruits and vegetables in all ELAN countries [37].

The consumption of foods considered healthy such as vegetables, fruits and dairy products occurred on a daily basis in the group of subjects classified with a healthy diet. Despite this, this same group showed a higher proportion of daily fat intake. No daily consumption of meats, sausages or sweets was found. 

Our sampling strategy resulted in a greater participation of women (81.6%) than men (18.4%), possibly reflecting the feminization that has been taking place in different countries; in Chile, the number of elderly women almost doubles that of the male population of the same age [42], which could be due to the greater longevity of women [43]. Also, women are usually more willing than men to participate in this kind of study [44]. 

The mean BMI of participants was 31 kg/m^2^, with only 25.8% of the population in a normal nutritional state. A total of 31.1% were overweight, 40.4% had obesity and 2.6% had low weight. These results are consistent with those obtained with other studies carried out in older people in Latin America [45,46,47,48,49]. The redistribution of subcutaneous fat with accumulation in the abdominal region, alongside the decrease in height renders the BMI an imprecise estimator of nutritional status in the elderly [50]. This situation is reflected in the waist circumference in our sample of patients, where 76.3% met the cardiovascular risk cutoff (79.1% of women and 44.9% of men). Even when waist circumference was not associated with quality of diet, cardiovascular risk determined by WC was associated with waist circumference in the logistic regression model, which is in line with findings by Tourlouki et al. [51]. In the Poisson regression model, cardiovascular risk was not associated with quality of diet. 

Regarding other clinical characteristics, less than half of the sample reported meeting the recommended hours of sleep, while most were not physically active. Polypharmacy (consumption > 5 drugs) was reported in 30.3%. None of these variables were associated with the quality of diet in our sample.

The strengths of our study include the evaluation of multiple components that are known to affect quality of diet in diverse populations, through the application of surveys which have been validated in Hispanic populations. Furthermore, there was a good response rate to this survey from inhabitants of the commune. Lastly, our study shows that structural health determinants like socioeconomic status may not fully explain decreased quality of diet in the elderly. Although this would need to be confirmed in future studies, it signals that it may be feasible to improve quality of diet in the elderly by tackling modifiable factors like food insecurity through community-level interventions.

The limitations of our study include that we collected self-reported data which may convey subjective recall bias. Furthermore, these associations were derived form a cross-sectional study design, which is one reason why more robust longitudinal studies could aim to characterize the directionality of these associations. Also, most participants reported a quality of diet in need of improvement, with few having an unhealthy diet HEI score, which limited our ability to fully characterize associations towards the unhealthy diet category. Another limitation is that we used the food consumption frequency questionnaire instead of other methods that are used to obtain further information, like the estimation of nutrient intake (i.e., 24-h dietary recall). Furthermore, we did not collect working status in our survey, which could be an important determinant of socioeconomic status in the elderly, especially since only 20.3% of people older than 65 years are active workers in Chile [52]. Lastly, our study may have limited generalizability to men, since most participants included in the survey were women.

## 5. Conclusions

In this sample of older adults from a southern Chilean commune, suboptimal quality of diet was frequent (85.9%), with most participants having a diet in need of improvement. Food insecurity and cardiovascular risk (determined according to waist circumference measurement) were associated with lower quality of diet categories (unhealthy diet and diet in need of improvement). Nonetheless, a direct association of these variables with the category of unhealthy diet was not confirmed with Poisson regression analyses, which could be explained by the low number of subjects in this category (n = 8, 2.1%). Other modifiable factors like physical activity, hours of sleep, and polypharmacy were not associated with lower quality of diet categories. Socioeconomic status, which is a structural health determinant, was not associated with decreased quality of diet. This suggest that structural health determinants are possibly not the strongest determinants of quality of diet in the elderly. Studies in the future could seek to better characterize these associations through longitudinal study designs. In the future, evidence from such studies could allow to design interventions aiming to reduce food insecurity, which could possibly serve to improve the quality of diet in older adults. 

## Figures and Tables

**Table 1 nutrients-14-00036-t001:** Descriptive characteristics of participants.

Name of Variable	Total Sample n = 376
Age, years *	73.5 (6.9)
Sex, n (%) **	
Women	307 (81.6)
Man	69 (18.4)
Weight, kg *	73.9 (12.9)
Height, m *	1.5 (0.1)
BMI, Kg/m^2^ *	31.3 (5.2)
BMI by classification, n (%) **	
<21.9 kg/m^2^	5 (1.3)
22–26.9 kg/m^2^	77 (20.5)
27–29.9 kg/m^2^	80 (21.3)
≥30 kg/m^2^	212 (56.4)
Waist circumference, cm *	99.6 (12.2)
Cardiovascular risk, n (%) **	287 (76.3)
Food Safety Scale, score *	0.9 (1.8)
Food Safety Scale Categories, n (%) **	
Security	247 (65.7)
Mild Insecurity	90 (23.9)
Moderate Insecurity	23 (6.1)
Severe Insecurity	12 (3.2)
Number of people in the household *	2.4 (1.3)
Median household income per month, n (%) **	
$158 USD	151 (40.2)
$307 USD	142 (37.8)
$503 USD	65 (17.3)
$810 USD	11 (2.9)
$1374 USD	4 (1.1)
$2070 USD	1 (0.3)
$4386 USD	2 (0.5)
Socioeconomic level, n (%) **	
Upper class	0 (0)
Wealthy Middle Class	2 (0.5)
Emerging Middle class	0 (0)
Typical Middle Class	13 (3.5)
Medium-low class	51 (13.6)
Vulnerable	161 (42.8)
Poor	149 (39.6)
Number of drugs, n (%) **	
0	31 (8.2)
1 to 2	88 (23.4)
3 to 5	143 (38.0)
6 to 8	67 (17.8)
>8	47 (12.5)
Hours of sleep per day, n (%) **	
8 to 10	164 (43.6)
5 to 7	171 (45.5)
<5	41 (10.9)
Physical activity, n (%) **	
Physically active	86 (22.9)
Irregulary active	124 (33.0)
Sedentary	166 (44.1)
Healthy Eating Index, score *	79.1 (10.1)
Healthy Eating Index Categories **	
Healthy	53 (14.1)
Needs improvement	315 (83.8)
Unhealthy	8 (2.1)

Data are presented as mean (SD) * or frequency (%) ** when specified. BMI: body mass index; kg: kilograms; m: meters; USD: United States dollars.

**Table 2 nutrients-14-00036-t002:** Comparisons of demographic characteristics according to quality of diet groups.

	Healthy Eating Index Categories			
	Healthy	Needs Improvement	Unhealthy	*p* Value
Number of people in the household *	2.4 (1.4)	2.5 (1.3)	2.3 (1.4)	0.914 ^a^
Age, years *	75.1 (7.3)	73.2 (6.9)	75.3 (4.5)	0.140 ^b^
Sex, n (%) **				
Women	44 (83.0)	257 (81.6)	6 (1.9)	0.794 ^c^
Man	9 (17.0)	58 (18.4)	2 (0.6)	
Weight, kg *	74.3 (13.1)	73.8 (12.9)	73.5 (13.0)	0.973 ^b^
Height, m *	1.5 (0.1)	1.5 (0.1)	1.6 (0.1)	0.437 ^b^
BMI, kg/m^2^ *	31.1 (5.5)	31.3 (5.2)	30.1 (5.3)	0.789 ^b^
BMI by classification, n (%) **				
<21.9	0 (0.0)	7 (2.2)	0 (0.0)	0.869 ^c^
22–26.9	13 (24.5)	62 (19.7)	2 (0.6)	
27–29.9	13 (24.5)	63 (20.0)	4 (1.3)	
≥30	27 (50.9)	183 (58.1)	2 (0.6)	
Obesity, n (%) **	27 (50.9)	183 (58.1)	2 (0.6)	0.920 ^c^
Waist circumference, cm *	97.7 (12.0)	99.9 (12.3)	99.5 (11.5)	0.488 ^b^
Cardiovascular risk, n (%) **	34 (64.2)	247 (78.4)	6 (1.9)	0.045 ^c^
Food insecurity, n (%) **	9 (17.0)	118 (37.5)	2 (0.6)	0.023 ^c^
Socioeconomic level, n (%) **				
Wealthy Middle Class	1 (1.9)	1 (0.3)	0 (0.0)	0.794 ^c^
Typical Middle Class	4 (7.5)	9 (2.9)	0 (0.0)	
Medium-low class	4 (7.5)	46 (14.6)	1 (0.3)	
Vulnerable	19 (35.8)	139 (44.1)	3 (1.0)	
Poor	25 (47.2)	120 (38.1)	4 (1.3)	
Number of drugs, n (%) **				
1 to 2	11 (20.8)	72 (22.9)	5 (1.6)	0.110 ^c^
3 to 5	19 (35.8)	123 (39.0)	1 (0.3)	
6 a 8	10 (18.9)	57 (18.1)	0 (0.0)	
>8	9 (17.0)	37 (11.7)	1 (0.3)	
Hours of sleep, n (%) **				
8 to 10	23 (43.4)	137 (43.5)	4 (1.3)	0.884 ^c^
5 to 7	25 (47.2)	142 (45.1)	4 (1.3)	
<5	5 (9.4)	36 (11.4)	0 (0.0)	
Physical activity, n (%) **				
Physically active	12 (22.6)	72 (22.9)	2 (0.6)	0.541 ^c^
Irregulary active	14 (26.4)	109 (34.6)	1 (0.3)	
Sedentary	27 (50.9)	134 (42.5)	5 (1.6)	

Data are presented as mean (SD) * or frequency (%) **. Data were compared by: a: ANOVA one-way, b: Welch’s ANOVA, c: chi-square for trend. BMI: body mass index; kg: kilograms; m: meters; USD: United States dollars.

**Table 3 nutrients-14-00036-t003:** Comparisons of food frequencies between the healthy eating index categories.

	Healthy Eating Index Classification				
	Total	Healthy	Needs Improvement	Unhealthy	*p* Value
Cereal consumption frequency					
Never or almost never	2 (0.5)	0 (0.0)	2 (0.6)	0 (0.0)	0.031
Occasional	11 (2.9)	0 (0.0)	10 (3.2)	1 (12.5)	
Weekly	40 (10.6)	5 (9.4)	32 (10.2)	3 (37.5)	
Daily consumption	323 (85.9)	48 (90.6)	271 (86.0)	4 (50.0)	
Vegetable consumption frequency					
Never or almost never	4 (1.1)	0 (0.0)	3 (1.0)	1 (12.5)	<0.001
Occasional	16 (4.3)	1 (1.9)	12 (3.8)	3 (37.5)	
Weekly	75 (19.9)	0 (0.0)	72 (22.9)	3 (37.5)	
Daily consumption	281 (74.8)	52 (98.1)	228 (72.4)	1 (12.5)	
Frequency of fruit consumption					
Never or almost never	6 (1.6)	0 (0.0)	5 (1.6)	1 (12.5)	<0.001
Occasional	30 (8.0)	0 (0.0)	26 (8.3)	4 (50.0)	
Weekly	95 (25.3)	4 (7.5)	90 (28.6)	1 (12.5)	
Daily consumption	245 (65.2)	49 (92.5)	194 (61.6)	2 (25.0)	
Frequency milk products consumption					
Never or almost never	26 (6.9)	0 (0.0)	24 (7.6)	2 (25.0)	<0.001
Occasional	51 (13.6)	1 (1.9)	47 (14.9)	3 (37.5)	
Weekly	123 (32.7)	14 (26.4)	106 (33.7)	3 (37.5)	
Daily consumption	176 (46.8)	38 (71.7)	138 (43.8)	0 (0.0)	
Fat consumption frequency					
Never or almost never	30 (8.0)	0 (0.0)	27 (8.6)	3 (37.5)	<0.001
Occasional	39 (10.4)	0 (0.0)	36 (11.4)	3 (37.5)	
Weekly	44 (11.7)	1 (1.9)	43 (13.7)	0 (0.0)	
Daily consumption	263 (69.9)	52 (98.1)	209 (66.3)	2 (25.0)	
Frequency of meat consumption					
Never or almost never	4 (1.1)	0 (0.0)	4 (1.3)	0 (0.0)	0.671
Occasional	47 (12.5)	3 (5.7)	41 (13.0)	3 (37.5)	
Weekly	277 (73.7)	50 (94.3)	224 (71.1)	3 (37.5)	
Daily consumption	48 (12.8)	0 (0.0)	46 (14.6)	2 (25.0)	
Legume consumption frequency					
Never or almost never	7 (1.9)	0 (0.0)	6 (1.9)	1 (12.5)	<0.001
Occasional	33 (8.8)	0 (0.0)	28 (8.9)	5 (62.5)	
Weekly	328 (87.2)	53 (100.0)	273 (86.7)	2 (25.0)	
Daily consumption	8 (2.1)	0 (0.0)	8 (2.5)	0 (0.0)	
Sausage consumption frequency					
Never or almost never	214 (56.9)	43 (81.1)	169 (53.7)	2 (25.0)	<0.001
Occasional	121 (32.2)	10 (18.9)	107 (34.0)	4 (50.0)	
Weekly	37 (9.8)	0 (0.0)	36 (11.4)	1 (12.5)	
Daily consumption	4 (1.1)	0 (0.0)	3 (1.0)	1 (12.5)	
Frequency of consumption of sweets					
Never or almost never	101 (26.9)	35 (66.0)	66 (21.0)	0 (0.0	<0.001
Occasional	195 (51.9)	17 (32.1)	174 (55.2)	4 (50.0	
Weekly	61 (16.2)	1 (1.9)	57 (18.1)	3 (37.5	
Daily consumption	19 (5.1)	0 (0.0)	18 (5.7)	1 (12.5	

Data are presented as frequency and percentage. Data were compared by Chi-square for trend.

**Table 4 nutrients-14-00036-t004:** Comparisons of food portion consumption between healthy eating index categories.

Healthy Eating Index Classification					
	Total	Healthy	Needs Improvement	Unhealthy	*p* Value
Cereals servings					
1 to 3 servings	337 (89.6)	50 (94.3)	279 (88.6)	8 (100)	0.418
3 to 5 servings	36 (9.6)	3 (5.7)	33 (10.5)	0 (0.0)	
>5 servings	1 (0.3)	0 (0.0)	1 (0.3)	0 (0.0)	
Does not consume	2 (0.5)	0 (0.0)	2 (0.6)	0 (0.0)	
Vegetables servings					
1 to 3 servings	361 (96)	49 (92.5)	304 (96.5)	8 (100)	0.530
3 to 5 servings	12 (3.2)	4 (7.5)	8 (2.5)	0 (0.0)	
>5 servings	0 (0.0)	0 (0.0)	0 (0.0)	0 (0.0)	
Does not consume	3 (0.8)	0 (0.0)	3 (1)	0 (0.0)	
Fruits servings					
1 to 3 servings	303 (80.6)	46 (86.8)	251 (79.7)	6 (75)	0.107
3 to 5 servings	66 (17.6)	6 (11.3)	59 (18.7)	1 (12.5)	
>5 servings	3 (0.8)	1 (1.9)	2 (0.6)	0 (0.0)	
Does not consume	4 (1.1)	0 (0.0)	3 (1)	1 (12.5)	
Milk products servings					
1 to 3 servings	349 (92.8)	52 (98.1)	290 (92.1)	7 (87.5)	0.050
3 to 5 servings	10 (2.7)	1 (1.9)	9 (2.9)	0 (0.0)	
>5 servings	1 (0.3)	0 (0.0)	1 (0.3)	0 (0.0)	
Does not consume	16 (4.3)	0 (0.0)	15 (4.8)	1 (12.5)	
Fat servings					
1 to 3 servings	348 (98)	53 (100)	291 (98.0)	6 (75.0)	0.131
3 to 5 servings	6 (1.7)	0 (0.0)	5 (1.7)	2 (25.0)	
>5 servings	0 (0.0)	0 (0.0)	0 (0.0)	0 (0.0)	
Does not consume	1 (0.3)	0 (0.0)	1 (0.3)	0 (0.0)	
Meat servings					
1 to 3 servings	309 (82.2)	49 (92.5)	252 (80.0)	8 (100)	0.136
3 to 5 servings	57 (15.2)	4 (7.5)	53 (16.8)	0 (0.0)	
>5 servings	8 (2.1)	0 (0.0)	8 (2.5)	0 (0.0)	
Does not consume	2 (0.5)	0 (0.0)	2 (0.6)	0 (0.0)	
Legumes servings					
1 to 3 servings	350 (93.1)	50 (94.3)	293 (93.0)	7 (87.5)	0.159
3 to 5 servings	20 (5.3)	3 (5.7)	17 (5.4)	0 (0.0)	
>5 servings	0 (0.0)	0 (0.0)	0 (0.0)	0 (0.0)	
Does not consume	6 (1.6)	0 (0.0)	5 (1.6)	1 (12.5)	
Sausages servings					
1 to 3 servings	220 (58.5)	22 (41.5)	190 (60.3)	8 (100)	0.001
3 to 5 servings	1 (0.3)	0 (0.0)	1 (0.3)	0 (0.0)	
>5 servings	0 (0.0)	0 (0.0)	0 (0.0)	0 (0.0)	
Does not consume	155 (41.2)	31 (58.5)	124 (39.4)	0 (0.0)	
Sweets servings					
1 to 3 servings	293 (95.4)	29 (96.7)	256 (95.2)	8 (100)	0.572
3 to 5 servings	12 (3.9)	0 (0.0)	12 (4.5)	0 (0.0)	
>5 servings	2 (0.7)	1 (3.3)	1 (0.4)	0 (0.0)	
Does not consume	0 (0.0)	0 (0.0)	0 (0.0)	0 (0.0)	

Data are presented as frequency and percentage. Data were compared by chi-square for trend.

**Table 5 nutrients-14-00036-t005:** Results of the ordinal logistic regression analyses to determine the association between demographic and lifestyle variables with quality of diet.

	β	SE	95% CI	*p* Value
Sex (male)	0.162	0.371	−0.560–0.883	0.672
Age (years)	−0.031	0.020	−0.071–0.011	0.160
BMI (points)	−0.001	0.027	−0.053–0.051	0.967
Waist circumference (cm)	0.013	0.012	−0.010–0.036	0.271
Cardiovascular risk				
Absent	Reference			
Present	0.631	0.309	0.026–1.236	0.041
Food Safety Scale (score)	0.190	0.092	0.010–0.370	0.038
Vulnerable or poor socioeconomic status				
Absent	Reference			
Present	0.007	0.985	−0.712–0.725	0.985
Obesity (BMI ≥ 30)				
Absent	Reference			
Present	0.053	0.281	−0.498–0.604	0.851
Food insecurity				
Absent	Reference			
Present	0.782	0.326	0.143–1.421	0.017
Pharmaceuticals				
1 to 2	Reference			
3 to 5	−0.729	0.488	−1.686–0.228	0.135
6 to 8	−0.574	0.453	−1.463–0.315	0.206
>8	−0.421	0.393	−1.192–0.350	0.284
Hours of Sleep				
8 to 10	Reference			
5 to 7	−0.023	0.474	−0.951–0.906	0.962
<5	−0.049	0.296	−0.629–0.531	0.868
Physical Activity				
Physically active	Reference			
Irregulary active	−0.122	0.358	−0.824–0.581	0.734
Sedentary	0.097	0.386	−0.660–0.854	0.802

β: regression coefficient, SE: standard error, 95% CI: 95% confidence interval, BMI: body mass index.

**Table 6 nutrients-14-00036-t006:** Results of Poisson regression to determine the association between demographic and lifestyle variables with quality of diet.

	β	SE	PR	95% CI	*p* Value
Sex (male)	0.811	0.866	2.250	0.412–12.284	0.349
Age (years)	0.003	0.060	1.003	0.892–1.127	0.961
BMI (points)	−0.152	0.095	0.859	0.713–1.036	0.111
Waist circumference (cm)	−0.031	0.038	0.970	0.901–1.044	0.414
Cardiovascular risk					
Absent	Reference				
Present	−0.573	0.866	0.564	0.103–3.079	0.508
Food Safety Scale, score	0.057	0.201	1.059	0.715–1.568	0.776
Vulnerable or poor socioeconomic status					
Absent	Reference				
Present	0.125	1.095	1.133	0.132–9.699	0.909
Obesity (BMI ≥ 30)					
Absent	Reference				
Present	−1.923	1.095	0.146	0.017–1.251	0.079
Food insecurity					
Absent	Reference				
Present	−0.057	0.866	0.945	0.173–5.159	0.948
Hours of Sleep					
8 to 10	Reference				
5 to 7	−0.713	0.866	0.490	0.090–2.676	0.410
<5	Not estimable				
Physical Activity					
Physically active	Reference				
Irregulary active	0.806	1.118	2.240	0.250–20.041	0.471
Sedentary	−0.242	1.414	0.785	0.049–12.551	0.864

β: regression coefficient, SE: standard error, 95% CI: 95% confidence interval, PR: prevalence ratio, BMI: body mass index.

## Data Availability

The data presented in this study are available on request from the corresponding author.

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
