# Peer review of "Food Insecurity Is Associated with the Quality of Diet of Non-Institutionalized Older Adults from a Southern Chilean Commune: A Cross-Sectional Study"

_nutrients, 2021, doi:10.3390/nu14010036_

Round 1

Reviewer 1 Report

It was my pleasure to read the paper since I found it interesting to read that. However, I made a few comments that can be addressed to improve the article.

Line 22: Delete (both women and men), rather provide % of men and women

Lines 33-37: Rephrase these sentences, same in the conclusion

Line 79: By any chance, do you know any prevalence of food insecurity among older adults in Chilean communes? If not, then it is OK.

Data collection and measurements: make a separate heading for Sleep and Physical activity 

2.2.2.. Food insecurity assessment: binary category (food security and food insecurity) for regression analysis??

Tables 3, 4 and 5: use * or bold P values those are statistically significant

Table 5: not sure about classification/ label for obesity and food insecurity (in terms of reference group)?

Discussion: would like to see justification association between poverty and food insecurity, and quality of diet

Limitations: (lines 310-316): better to add -collected self-reported data and thus, may have subjective recall bias.

Thank you.

Author Response

Dear editors and reviewers,

We would like to thank you for your efforts and time spent in reviewing our manuscript. Your recommendations have allowed us to significantly improve our manuscript.

These are our individual responses to your comments:

Reviewer 1

It was my pleasure to read the paper since I found it interesting to read that. However, I made a few comments that can be addressed to improve the article.

Line 22: Delete (both women and men), rather provide % of men and women

R: Thank you, we have made this modification within the abstract.

Lines 33-37: Rephrase these sentences, same in the conclusion

R: We have now rephrased these sentences in both the abstract and conclusion section. We hope that this has brought more clarity.

Line 79: By any chance, do you know any prevalence of food insecurity among older adults in Chilean communes? If not, then it is OK.

R: This specific reference cited in the manuscript refers to the prevalence of food insecurity in older adults from Chilean communes (40.4%). We have now further clarified it in the manuscript. Tu further comment on your query, there have been more recent studies evaluating food insecurity in the context of the second wave of the COVID-19 pandemic, reporting a prevalence of 37% in older adults during the second wave of COVID-19 in Chile. This prevalence is similar to the one we initially cited in our manuscript (40.4%) which also referred to older adults in Chile as priorly stated.

The reference for the most recent survey in the context of COVID-19 is: Herrera Ponce Soledad, Elgueta Rosas Raúl, Fernández Lorca M.Beatriz, Giacoman Hérnandez Claudia, Leal Valenzuela Daniella, Rubio Acuňa Miriam, Marshall De la Maza Pío & Bustamante Palma Felipe. (2021). Calidad de vida de las personas mayores chilenas durante la pandemia Covid-19. Agencia Nacional de Investigación y Desarrollo del Gobierno de Chile.

Data collection and measurements: make a separate heading for Sleep and Physical activity

R: We have modified this as suggested.

2.2.2.. Food insecurity assessment: binary category (food security and food insecurity) for regression analysis??

R: The variable of food insecurity for this analysis was created by merging categories from the Food Safety Scale variable. We considered all three categories which reflect any degree of food insecurity (mild, moderate, and severe) as a single one, whereas the category of Food Security was kept as is (the absence of food insecurity). This dichotomic variable was used in both regression analyses. We have further described this in our manuscript in the materials and methods section to give more clarity on how variables were managed for these analyses. We have also detailed this in tables 5 and 6 by mentioning that food insecurity was modelled as absent vs present.

Tables 3, 4 and 5: use * or bold P values those are statistically significant

R: Thank you. We have modified this as suggested.

Table 5: not sure about classification/ label for obesity and food insecurity (in terms of reference group)?

R: The obesity and food insecurity variables were used in the logistic regression analyses as dichotomous variables (present versus absent). We considered a BMI ≥30 as the presence of obesity, while the reference category was a BMI <30 which was considered as not having obesity. For food insecurity, we have discussed in the prior query how this variable was dichotomized. For both variables, tables 5 and 6 now reflect in a better way how these variables were managed.  

Discussion: would like to see justification association between poverty and food insecurity, and quality of diet

R: Thank you for recommending this. We have added a few sentences in our discussion to provide possible explanations of these associations in our sample of older adults.

Limitations: (lines 310-316): better to add -collected self-reported data and thus, may have subjective recall bias.

R: Thank you for noting this. We have added it to the limitations of our study.

Reviewer 2 Report

Dear Authors,

The manuscript (nutrients-1509878) submitted for review is very interesting and well written. I recommend the article to make minor revision and adding new information, which in my opinion is missing, and which will allow the readers to better understand this study.

Authors, Please note and address the following comments:

Title of the manuscript In my opinion, the title of the manuscript should be changed. Now, it is a bit complicated, but I have no idea how to change it. 

Material and methods

Who collected data in the study? The authors only mentioned the people who validated the questionnaire. It is especially worth mentioning who made the anthropometric measurements.

Can the authors supplement the survey validation data? Was it validation or was it just preliminary research? Authors, please add additional information.

No information about the number of questions in the questionnaire.

Results

Did the data from the pilot study also be included in the analysis of the results?

Table 1 – The second column of Table 1 is called – 'Statistical'. In my opinion, it should rather be the Number (SD) of participants.

Lines 209 – below Table 1 should be Frequency.

I have a question. Are there any limitations of this manuscript? If yes, it is a good idea to write about it.

Conclusion

What are the practical and theoretical implications of the research?

Do the authors have any suggestions for directions for further research?

References

Most of the references (39 from 48) come from the last 10 years. References are cited according to journal rules.

Despite my comments, I am pleased to recommend this manuscript . I believe it addresses an important area of research in an international context. The article needs improvement. I hope these comments will help the authors in their work to improve the manuscript.

Reviewer

Author Response

Dear editors and reviewers,

We would like to thank you for your efforts and time spent in reviewing our manuscript. Your recommendations have allowed us to significantly improve our manuscript.

These are our individual responses to your comments:

Reviewer 2

Dear Authors,

The manuscript (nutrients-1509878) submitted for review is very interesting and well written. I recommend the article to make minor revision and adding new information, which in my opinion is missing, and which will allow the readers to better understand this study.

Authors, Please note and address the following comments:

Title of the manuscript -  In my opinion, the title of the manuscript should be changed. Now, it is a bit complicated, but I have no idea how to change it.

R: Thank you for suggesting this. We have modified the title to focus on the main variable which was associated with quality of diet in our study. The new title is Food insecurity is associated with the quality of diet of non-institutionalized older adults from a southern Chilean commune: A cross-sectional study

Material and methods

Who collected data in the study? The authors only mentioned the people who validated the questionnaire. It is especially worth mentioning who made the anthropometric measurements.

R: We have clarified this as follows in the manuscript: All data were collected by a team composed of Nutritional Scientists and last-year students of the bachelor’s degree in nutrition and dietetics, who received training on standardization of anthropometric measurement collection and surveying; a Standardized Operating Procedures protocol was built for this.

Can the authors supplement the survey validation data? Was it validation or was it just preliminary research? Authors, please add additional information.

R: This was only a validation. We have added more detailed information of all the procedures that this validation involved as follows: The survey was validated by 10 experts in nutrition and public health in a pilot test, which consisted of invitation of 30 persons older than 60 years from the commune, who were subsequently ineligible to be included in the sample of the main study, that provided their written informed consent to participate in the pilot study to validate the survey. The procedures of the validation involved linguistic and cultural adaptations, and viability by assessing the time employed in the application of the survey, easiness of the format, and brevity and clarity of the questions. Items were adapted to include language understandable by the elderly, as well as to improve the way of delivering questions by interviewers, as well as registration and codification of responses.

No information about the number of questions in the questionnaire.

R: The survey consisted of a total of 50 items within 7 sections. We have clarified this in the manuscript. Also, we are now providing the full survey as Supplementary Materials.

Results

Did the data from the pilot study also be included in the analysis of the results?

R: Data from participants of the pilot study were not included for analyses. We have further clarified this in our methods.

Table 1 – The second column of Table 1 is called – 'Statistical'. In my opinion, it should rather be the Number (SD) of participants.

R: Thank you for noting this error. We have corrected it as suggested.

Lines 209 – below Table 1 should be Frequency.

R: Thank you, we have also corrected this.

I have a question. Are there any limitations of this manuscript? If yes, it is a good idea to write about it.

R: We recognize that there are important limitations to our study. We have recognized limitations of our study in the last paragraph, immediately before the “Conclusion” section. These are the limitations that we have written:

Limitations of our study include that we collected self-reported data which may convey subjective recall bias. Furthermore, these associations are derived form a cross-sectional study design, reason why more robust longitudinal studies could aim to characterize directionality of these associations. Also, most participants reported a quality of diet in need of improvement, with few having an unhealthy diet HEI score, which limited our ability to fully characterize associations towards the unhealthy diet category. Lastly, our study may have limited generalizability to men since most participants included in the survey were women.

Conclusion

What are the practical and theoretical implications of the research?

R: Thank you, we have added these implications in the conclusion. Since this was a cross-sectional study, we tried to be extra cautious with our conclusions since our study should be interpreted as hypothesis generating to promote more research on the topic, but it should not be interpreted as definitive.

Do the authors have any suggestions for directions for further research?

R: Thank you for suggesting this. We have also added our perspectives in the conclusion section.

References

Most of the references (39 from 48) come from the last 10 years. References are cited according to journal rules.

R: Thank you.

Despite my comments, I am pleased to recommend this manuscript . I believe it addresses an important area of research in an international context. The article needs improvement. I hope these comments will help the authors in their work to improve the manuscript.